# Mental operations in rhythm: Motor-to-sensory transformation mediates imagined singing

Yanzhu Li[1,2], Huan Luo[3], Xing Tian[1,2]*

1 New York University Shanghai, Shanghai, China, 2 NYU-ECNU Institute of Brain and Cognitive Science at NYU Shanghai, Shanghai, China, 3 Peking University, Beijing, China

* xing.tian@nyu.edu

**Data Availability Statement:** All MEG data files are available from the OSF database (https://osf.io/MC8WD/).

**Funding:** This study was supported by the National Natural Science Foundation of China (NSFC) 31871131, the Major Program of Science and

## Abstract

What enables the mental activities of thinking verbally or humming in our mind? We hypothesized that the interaction between motor and sensory systems induces speech and melodic mental representations, and this motor-to-sensory transformation forms the neural basis that enables our verbal thinking and covert singing. Analogous with the neural entrainment to auditory stimuli, participants imagined singing lyrics of well-known songs rhythmically while their neural electromagnetic signals were recorded using magnetoencephalography (MEG). We found that when participants imagined singing the same song in similar durations across trials, the delta frequency band (1–3 Hz, similar to the rhythm of the songs) showed more consistent phase coherence across trials. This neural phase tracking of imagined singing was observed in a frontal-parietal-temporal network: the proposed motor-to-sensory transformation pathway, including the inferior frontal gyrus (IFG), insula (INS), premotor area, intraparietal sulcus (IPS), temporal-parietal junction (TPJ), primary auditory cortex (Heschl's gyrus [HG]), and superior temporal gyrus (STG) and sulcus (STS). These results suggest that neural responses can entrain the rhythm of mental activity. Moreover, the theta-band (4–8 Hz) phase coherence was localized in the auditory cortices. The mu (9–12 Hz) and beta (17–20 Hz) bands were observed in the right-lateralized sensorimotor systems that were consistent with the singing context. The gamma band was broadly manifested in the observed network. The coherent and frequency-specific activations in the motor-to-sensory transformation network mediate the internal construction of perceptual representations and form the foundation of neural computations for mental operations.

## Introduction

"What is this paper about?" You are probably asking this question in your mind. We think in a verbal form all the time in our everyday lives. Verbal thinking is one of the common mental operations that manifest as inner speech—a type of mental imagery induced by covert speaking [1–3]. Another related common mental phenomenon is the "earworm"—a piece of music that repeats in someone's mind or the involuntary action of humming a melody. What enables these mental operations that take the form of speech and singing?

Technology Commission of Shanghai Municipality (STCSM) 17JC1404104, and the Program of Introducing Talents of Discipline to Universities, Base B16018 to XT. The funders had no role in study design, data collection and analysis, decision to publish, or preparation of the manuscript.

**Competing interests:** The authors have declared that no competing interests exist.

**Abbreviations:** AG, angular gyrus; aSTG, anterior superior temporal gyrus; aSTS, anterior superior temporal sulcus; BCI, brain-computer interface; CFR, Code of Federal Regulations; dSPM, dynamic statistical parametric mapping; FFT, fast Fourier transform; FWHM, Full Width at Half Maximum; HG, Heschl's gyrus; ICA, independent component analysis; IFG, inferior frontal gyrus; INS, insula; IPS, intra-parietal sulcus; IRB, Institutional Review Board; ITC, inter-trial phase coherence; m&pSTS, middle and posterior superior temporal sulcus; MEG, magnetoencephalography; pINS, posterior insula; PO, parietal operculum; PreM, premotor cortex; RMS, root-mean-square; RT, reaction time; SMG, supramarginal gyrus; STG, superior temporal gyrus; STS, superior temporal sulcus; TPJ, temporal-parietal junction.

Neural evidence suggests that modality-specific cortical processes mediate covert operations of mental functions. For example, previous studies have demonstrated that mental imagery is mediated by neural activity in modality-specific cortices, such as the motor system for motor imagery [4, 5] and sensory systems for visual imagery [6, 7] and auditory imagery [8, 9].

Recently, the internal forward model has been proposed to link the motor and sensory systems [10]. The presupposition of the model is a motor-to-sensory transformation—a copy of motor command, termed "efference copy," is internally sent to sensory regions to estimate the perceptual consequence of actions [11–13]. This motor-to-sensory transformation has been demonstrated in speech production, learning, and control [14–18] and has been extended to speech imagery [3, 19–26]. This motor-to-sensory transformation has been suggested to be in a frontal-parietal-temporal network. Specifically, it has been assumed that the motor system in the frontal lobe simulates the motor action, while the sensory systems in the parietal and temporal lobes estimate the possible perceptual changes caused by the action [3, 22, 25]. Would the continuous simulation and estimation in the motor-to-sensory transformation network mediate the mental operations of covert singing and inner speech during verbal thinking [27]?

Thinking verbally and singing covertly are similar to speech because they all unfold over time. The analysis of time-series information in speech perception has been investigated using a neural entrainment paradigm. It has been demonstrated that neural responses can be temporally aligned to the frequency of acoustic features, such as speech envelopes [28, 29]. The neural responses can also entrain perceptual and cognitive constructs, such as syllabic information [30], music beats [31, 32], syntactic structures [33], and language formality structures [34]. That is, the frequency of neural responses can mirror the rate of internal representation derived from external stimulations. Would neural responses track the representations that are constructed without external stimulation, such as covert singing and inner speech during verbal thinking?

This study aims to use a rhythmic entrainment paradigm to investigate the neural mechanisms that mediate mental operations such as inner speech and covert singing. This "entrainment" is in a broader sense, as suggested by Obleser and Kayser [35]. We implemented a natural and rhythmic setting in which participants imagined singing lyrics of well-known songs by imitating the same songs heard before the imagery tasks (Fig 1A). A color change of the fixation cued the onset of imagery. The offset of imagery was indicated by a button press by participants. Unlike the frequency tracking of passive listening to an external stimulus that has a consistent rate across trials, the production rate in the active imagery task inevitably has temporal variance across trials. We used 2 approaches to deal with the temporal variation. First, the purpose of a musical context was to reduce the large temporal variability during imagery—participants would imagine singing at a more consistent rate compared to saying the same lyrics. Second, we took advantage of the remaining temporal variation among trials in imagery (Fig 1B, 1C and 1D). The variation in the duration of performing imagery tasks correlated with the temporal consistency of neural responses across trials. If the neural responses tracked the rate of mental operations, the phase coherence of neural responses would be different between 2 groups of imagery trials that have different durations (Fig 1E). According to our hypothesis that the motor-to-sensory transformation neural network mediates inner speech and covert singing, we predicted that the different degrees of neural tracking to the rate of mental operations would be observed in specific areas in the frontal, parietal, and temporal regions (Fig 1F), where the core computations for motor simulations and perceptual estimations in motor-to-sensory transformation have been indicated [3, 22, 25].

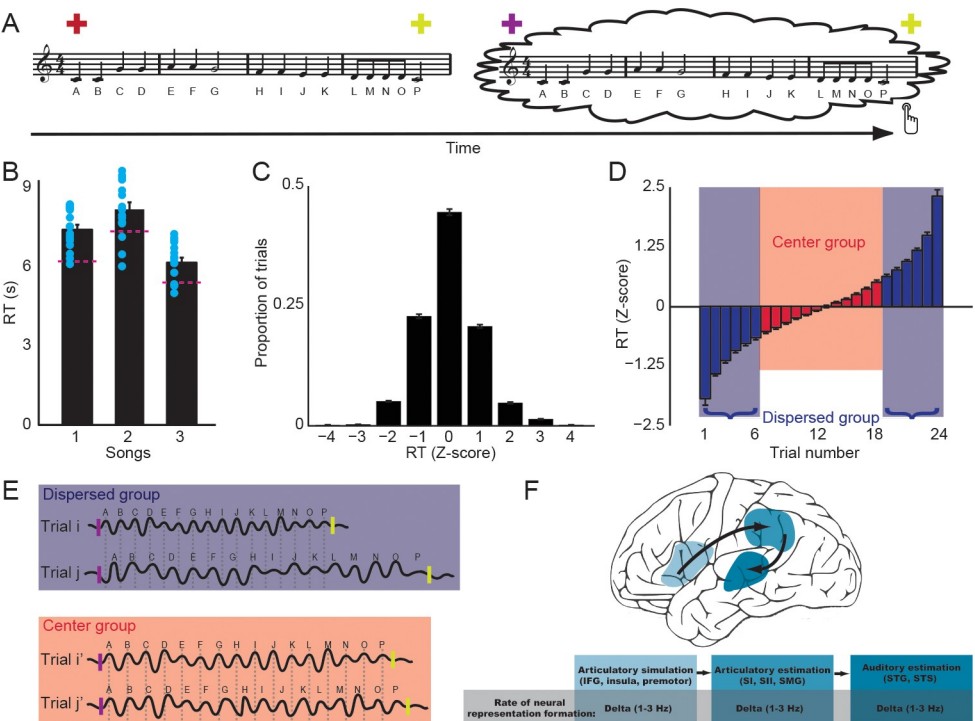

**Fig 1. Rhythmic entrainment of imagined singing and hypothesis of neural phase tracking.** (A) Experimental paradigm. According to the color of visual fixation, participants listened to the first sentence of 3 well-known songs, followed by imagined singing of the song they just heard (the Alphabet Song is used for illustration). Participants pressed a button to indicate the end of their imagery. (B) RT of imagined singing for the 3 songs. The red dashed lines indicate the duration of the 3 songs. The duration of imagined singing was longer than the preceding auditory stimuli. Each blue dot indicates an individual RT. (C) Distribution of imagined singing RT. The z-scores of RT followed a normal distribution, with about half of the trials within 2 standard deviations. (D) Grouping of imagined singing trials. Twenty-four trials of each song were sorted in ascending order based on their z-scores and separated into 2 groups. The 12 trials that were close to the mean RT were selected for the center group, whereas the other 12 trials that were further away from the mean RT were included in the dispersed group. (E) Hypothesis about neural phase coherence across trials of imagined singing. Schematic display of 2 trials in each group. The short bars indicate the beginning and end of a trial. The wave lines represent neural oscillations. The trials in the dispersed group have different durations, thus the temporal variance was large. The phase of neural oscillation that corresponded to the construction of syllabic representation during imagined singing did not align across trials. On the other hand, the temporal variance was small across trials in the center group, hence the phase of neural oscillation is more coherent across trials. (F) Hypothesis about phase coherence in the motor-to-sensory transformation network during imagined singing. The motor-to-sensory transformation was assumed to manifest in a frontal-parietal-temporal network, including the IFG, INS, premotor area, and SMA in the frontal lobe for simulating articulation; somatosensory areas (SI, SII), SMG, and its adjacent PO, AG, and TPJ in the parietal lobe for estimating somatosensory consequence; as well as the STG and STS with a possibility of extension to the HG in the temporal lobe for estimating auditory consequence. The more consistent phase coherence at the delta band (1–3 Hz)—the rate of imagined singing—was predicted to be observed in the motor-to-sensory transformation network. AG, angular gyrus; HG, Heschl's gyrus; IFG, inferior frontal gyrus; INS, insula; PO, parietal operculum; RT, reaction time; SI, primary somatosensory area; SII, secondary somatosensory area; SMA, supplementary motor area; SMG, supramarginal gyrus; STG, superior temporal gyrus; STS, superior temporal sulcus; TPJ, temporal-parietal junction.

## Results

The reaction time (RT) in the imagery condition suggested that the duration of imagined singing was longer than the duration of the auditory stimuli (Fig 1B) (one-sample $t$ test; for song 1, $t(15) = 6.04$, $p < 0.001$; for song 2, $t(15) = 2.64$, $p = 0.02$; for song 3, $t(15) = 2.82$, $p = 0.01$). A repeat measures one-way ANOVA did not reveal differences, in the increase of duration, among imagined singing of the 3 songs ($F(2) = 2.17$, $p = 0.126$). This suggested that the reduced speed in the imagery condition, presumably caused by the imagery task and motor

responses of button pressing, was consistent during imagery of all songs. The distribution of RT (Fig 1C) followed a normal distribution (chi-squared goodness-of-fit test, $\chi^2(4) = 2.30$, $p = 0.68$) and revealed that about half of the trials fall within ±1 standard deviations. Two groups of trials, based on the variation from the mean of RT, were formed for further analysis of the imagined singing magnetoencephalography (MEG) responses. For imagery of each song, 12 trials that were close to the mean RT were included in the center group, whereas the 12 trials that were further away from the mean RT were included in the dispersed group (Fig 1D).

We first examined the MEG responses in the temporal domain. The event-related responses that were time-locked to the onset of auditory stimuli revealed a clear peak and topography around 100 ms after the stimulus onset—the typical M100 auditory response (Fig 2A). More-over, in the imagery condition, a topographic pattern that was similar to the M100 auditory response was observed around a similar latency (Fig 1B), even though no external auditory stimulus was presented in the imagery condition. Both M100 responses demonstrated a canonical auditory response topography in which a dipole pattern over the temporal lobe region in each hemisphere was observed [e.g., 36]. This response topography contrasted with earlier visual responses in MEG that showed more posterior dipole patterns [e.g., 37, 38]. Therefore, the M100 responses in both the listening and imagery conditions were less likely to be induced by the fixation color changes. This similar event-related auditory response in the listening and imagery conditions was consistent with our previous findings [22] and suggested that auditory cortices were activated during the imagery condition.

No repetitive topographic patterns were observed in the time course of listening or imagery (Fig 1A and 1B), suggesting that tracking of the acoustic stream or the rate of imagery was not by the response magnitude. These results are consistent with previous studies that showed the absence of power coherence in speech tracking [29]. The lack of effects in response magnitude was probably due to repetition suppression [39]. Repetitions of similar processes decreased the response magnitude. Moreover, the temporal variance in imagery could further reduce the magnitude of evoked responses. Therefore, we further investigated the timing information and neural tracking of the induced responses in the spectral domain.

The MEG responses in the listening condition showed the neural tracking of rhythm in the songs. The phase-coherence analysis revealed that the significant differences ($p_{corr(FDR)} < 0.05$) between the inter-trial phase coherence (ITC) of the within-group and between-group were localized mostly in the primary auditory cortex (Heschl's gyrus [HG]) and its adjacent areas, including the left anterior superior temporal gyrus (aSTG) and sulcus (aSTS), and bilateral posterior insula (pINS) (Fig 2C). (See S1 Fig and S1 Table for precise anatomical locations.) These results suggest that auditory systems can reliably follow the rhythm of acoustic signals and demonstrate the validity and accuracy of source localization based on phase coherence.

For the imagery condition, the comparison between the ITC in the center group and dispersed group revealed 3 significant clusters ($p_{corr(FDR)} < 0.05$) in the frontal, parietal, and temporal regions (Fig 2D). Specifically, in the frontal region, more consistent phase coherence was observed in the inferior frontal gyrus (IFG), insular cortex (INS), and premotor cortex (PreM). In the parietal region, the differences were in the bilateral intra-parietal sulcus (IPS) and temporal-parietal junction (TPJ). In the temporal region, more consistent phase coherence was localized in the bilateral HG, left aSTG, and right middle and posterior superior temporal sulcus (m&pSTS). These phase-coherence results during imagined singing were observed in the proposed core computational regions of motor-to-sensory transformation (Fig 1F). These results suggest that neural responses track the dynamics of mental operations in the motor-based prediction pathway [3, 22, 23, 25].

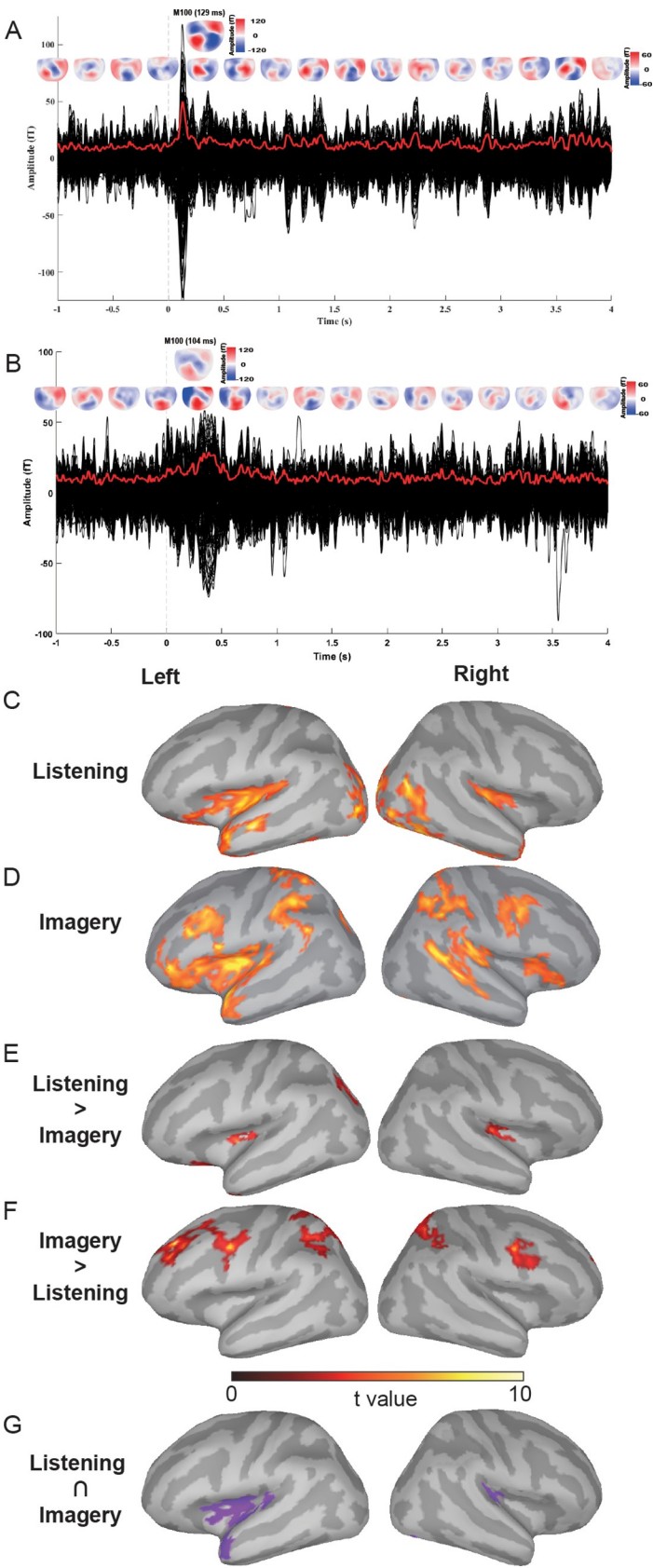

**Fig 2. MEG results of temporal responses and neural tracking in the delta band (1–3 Hz).** (A) Waveforms and topographies in the listening condition (the Alphabet Song). The vertical, dotted line at time 0 indicates the onset of the auditory stimuli. Each black line represents the waveform responses from a sensor. The red bold line represents the RMS waveform across all sensors. Topographies are plotted every 333 ms from −1,000 ms to 4,000 ms. A clear auditory onset event-related response (M100, the single topography in the upper row) was observed. (B) Waveforms and topographies in the imagery condition. Similar depicting form as in A. The vertical dotted line at time 0 indicates the onset of imagined singing. No repetitive patterns in topographies across the period. A similar event-related response in the range of M100 latency, as in the listening condition, was observed (the single topography in the upper row). (C) Phase-coherence results in the listening condition. Neural entrainment in the delta band was observed in the HG and its adjacent aSTG, aSTS, and pINS. (D) Phase-coherence results in the imagery condition. More consistent neural entrainment in the delta band was observed in the proposed motor-to-sensory network, including frontal areas (IFG, aINS, and premotor area), parietal areas (IPS and TPJ), and temporal areas (HG, aSTG, and m&pSTS). These observed cortical areas were consistent with the predicted frontal-parietal-temporal regions in Fig 1F. (E) Results of direct comparison between the listening and imagery conditions (Listening > Imagery). Greater phase coherence in listening than imagery was observed in bilateral HG and adjunct pINS. (F) Results of direct comparison between the listening and imagery conditions (Imagery > Listening). Greater phase coherence in listening than imagery was observed in premotor areas and IPS. (G) Results of conjunction analysis between the listening and imagery conditions. Overlapped significant phase coherence was observed in the bilateral primary auditory cortex, left pINS, and aSTG. The underlying data for this figure can be found at https://osf.io/mc8wd/. aINS, anterior insula; aSTG, anterior superior temporal gyrus; aSTS, anterior superior temporal sulcus; HG, Heschl's gyrus; IFG, inferior frontal gyrus; IPS, intra-parietal sulcus; MEG, magnetoencephalography; m&pSTS, middle and posterior superior temporal sulcus; pINS, posterior insula; RMS, root-mean-square; TPJ, temporal-parietal junction.

The reliability of the neural entrainment in listening and imagery was further tested using an extended period of 6-s data epochs (S2 Fig). The results were consistent with those obtained using 4-s epochs (Fig 2C). The entrainment was localized at the bilateral primary auditory cortex, aSTG, and INS in the listening condition, whereas localization of phase-coherence differences in the imagery condition was observed in the proposed frontal-parietal-temporal network, similar to the results in Fig 2D. Moreover, the analysis of four 3-s consecutive time bins revealed that the entrainment to imagery required time to be established, as the phase-coherence differences became more stable in the later time bins (S3 Fig).

The direct comparison between listening and imagery revealed stronger phase coherence in the HG and pINS in the listening condition compared with the imagery condition (Fig 2E) but greater phase coherence in the premotor areas and IPS in the imagery condition compared with the listening condition (Fig 2F). These results were consistent with common observations that perception has a more robust neural activation than imagery and additional frontal and parietal areas are engaged in speech imagery [e.g., 22]. More importantly, in the conjunction analysis, the significant phase coherence in listening and imagery conditions overlapped over the primary auditory cortex and extended to the posterior part of the INS and in the anterior part of the STG (Fig 2G). These overlaps suggest that the motor-to-sensory transformation during imagery can induce similar auditory representation as in perception [21].

To further explore the functional specificity of dynamic processing in the motor-to-sensory pathway, we performed the phase-coherence analysis in a broad range of frequency bands (Fig 3). We found that in the 4–8 Hz (theta) band, only the primary auditory cortex and anterior part of the STG and STS in the left hemisphere showed significant phase coherence. This focused theta-band activation in the auditory cortices—contrasted with the collaborative frontal-parietal-temporal network activation of the entire motor-to-sensory pathway in the delta band (Fig 1)—was consistent with the specific role of the theta band in auditory and speech processing [e.g., 40, 41].

A broad sensorimotor network in the right hemisphere was observed in 9–12 Hz (the alpha band, or the "mu" rhythm) and 16–20 Hz (the mid-beta band). These frequency bands also showed higher phase coherence in the inferior parts of the subcentral gyrus and opercular part of the IFG in the left hemisphere; the 16–20 Hz also extended to the PreM. The specific

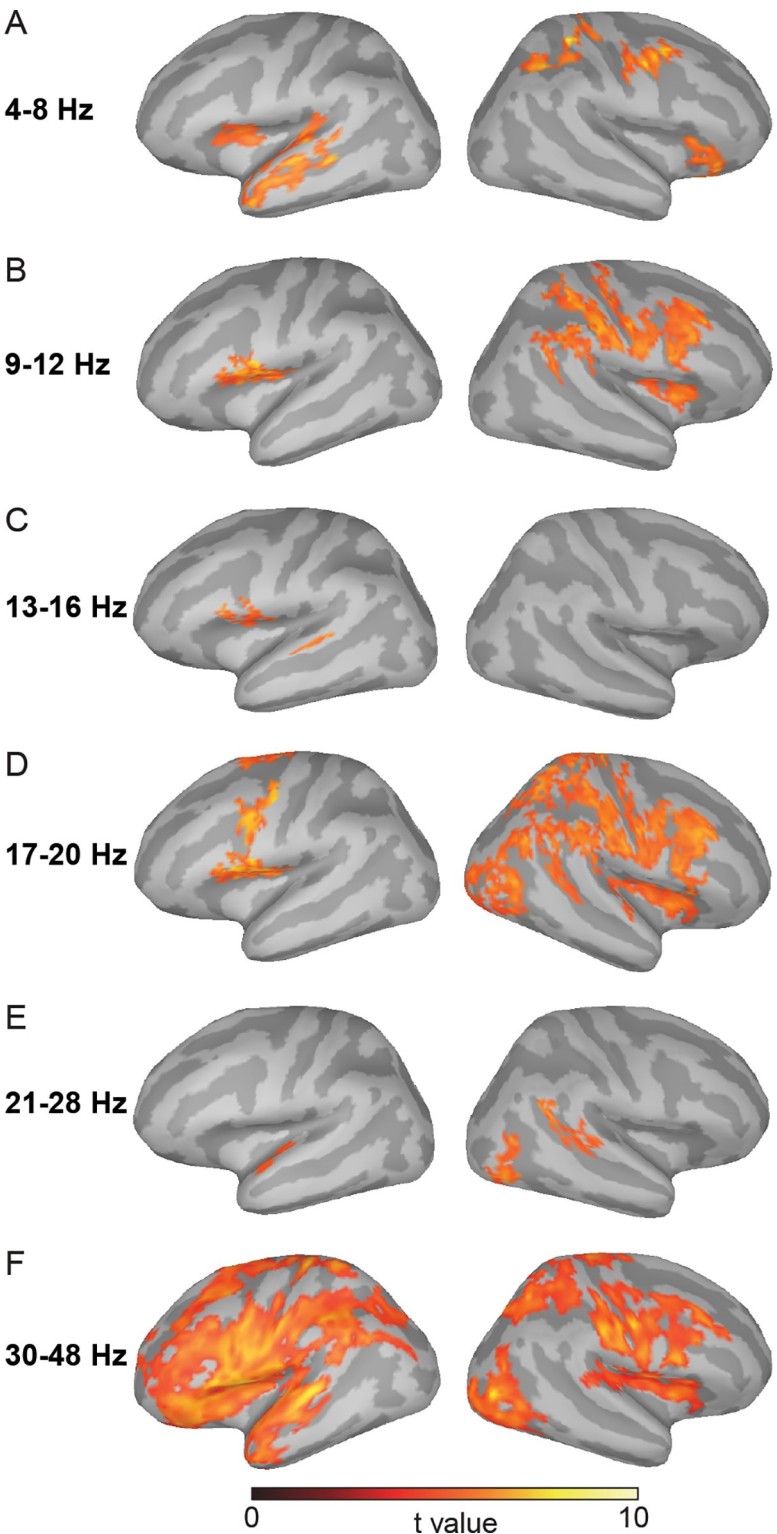

**Fig 3. Localization results of phase coherence in the imagery condition in the theta (4–8 Hz), mu (9–12 Hz), low-beta (13–16 Hz), mid-beta (17–20 Hz), high-beta (21–28 Hz), and low-gamma (30–48 Hz) bands.** (A) The phase coherence in the theta band was observed in the left primary auditory cortex, STG, and STS. (B) The phase coherence in the mu band was observed over the sensorimotor cortices in the right hemisphere, as well as in the inferior parts of the subcentral gyrus, and the opercular part of the IFG in the left hemisphere. (C) Only a small patch of subcentral gyrus, IFG, and mSTG was observed in the low-beta band. (D) The phase coherence in the mid-beta band was

observed in similar and broader right sensorimotor cortices as in the mu band in B. In the left hemisphere, the PreM was observed in addition to the observed areas in the mu band. (E) In the high-beta band, the phase coherence was observed in pSTG and pSTS in the right hemisphere and a small patch of aSTG in the left hemisphere. (F) Similar and broader networks, as observed in the neural tracking of the delta band in Fig 2D, were observed in the low-gamma band. The underlying data for this figure can be found at https://osf.io/mc8wd/. aSTG, anterior superior temporal gyrus; IFG, inferior frontal gyrus; mSTG, middle superior temporal gyrus; PreM, premotor cortex; pSTG, posterior superior temporal gyrus; pSTS, posterior superior temporal sulcus.

involvement of sensory and motor systems in the mu and beta bands were consistent with the alpha-beta synchronization in motor control and motor imagery [e.g., 42, 43], suggesting the specific dynamics for the motor simulation and somatosensory estimation in the motor-to-sensory transformation [3, 22, 25]. The right hemisphere dominance was consistent with imagery in a singing context [e.g., 44, 45, 46].

In the low-gamma band, greater synchronization was observed in the center group than that in the dispersed group over the entire motor and sensory systems that showed tracking of imagery in the delta band (Fig 2D). These results are consistent with the view of local computations for the gamma band [47, 48]. The distinctive neural involvement in the tracking rate of the delta—as well as in the theta, mu, beta, and gamma bands—collaboratively indicates the extent of the motor-to-sensory transformation network and the specific functional dynamics of each frequency component in this network.

## Discussion

We investigated the function and dynamics of neural networks that mediate the mental operations of inner speech and covert singing. With a rhythmic entrainment imagined singing paradigm, we found that frontal-parietal-temporal regions in the proposed motor-to-sensory network collaboratively synchronize at the rate of mental operations. Moreover, a double dissociation of operating frequencies and anatomical locations was found in the motor, somatosensory, and auditory cortices that distinctively relate to inner speech and covert singing. These results suggest that neural responses can entrain the rhythm of mental activity and the synchronized neural activity in the motor-to-sensory transformation network mediates mental operations.

Neural dynamics are crucial for understanding the computations that mediate cognitive functions [49, 50]. However, probing dynamics is difficult, especially the ones that mediate mental operations without external stimulations. The active nature of mental tasks inevitably causes temporal variations that undermine the foundations of methods for probing dynamics. For example, the phase-coherence analysis in the neural entrainment approach is well established for speech perception. However, the same analysis cannot be directly applied to investigate the neural tracking of speech imagery—the within-group would have too much temporal variance across trials, and therefore the phase coherence would be too small to detect. In this study, we overcame this obstacle by taking advantage of the temporal variance—separating trials based on behavioral timing (Fig 1D). This methodological advance, along with the use of naturalistic sounds and source localization, makes it possible to investigate the dynamics in neural networks that mediate mental operations.

Using the functional constraints of phase coherence, the observed frontal-parietal-temporal network during imagined singing (Fig 2D) was consistent with the proposed motor-to-sensory transformation network. The observed IFG, premotor, and INS in the frontal region during imagined singing have been demonstrated in articulatory preparation during overt speech [30, 51, 52] and covert speech [53, 54]. The responses in these frontal cortices were also consistent with findings in speech imagery, reflecting the function of motor simulation [25]. In the

parietal region, the observation of the IPS and TPJ—an area closed to the supramarginal gyrus (SMG), parietal operculum (PO), and angular gyrus (AG)—has also been suggested for sensorimotor integration and goal-directed prediction-based speech feedback control [55–58]. The activation of similar parietal areas of the TPJ, SMG, PO, and adjacent IPS was also observed during speech imagery, suggesting the possible functions for estimating somatosensory consequences of actions [22, 25].

The observations in the temporal region further support the motor-to-sensory transformation in covert singing and inner speech in verbal thinking. The similar evoked M100 responses in both listening and imagery conditions (Fig 2A and 2B) suggested that auditory representations can be established in both bottom-up and top-down manners. These auditory representations in imagery were probably induced by different processes from those mediating omission responses [e.g., 59], because the regular occurrence of an active task was required in this imagery study. Moreover, overlapped phase-coherence results were observed in the temporal regions of the primary and secondary auditory cortices between the listening and imagery conditions (Fig 2G). The STG was commonly observed during musical imagery [60], and the activation of auditory imagery has been shown to extend to the HG [8]. The observation of the HG during imagined singing in this study was consistent with the hypothesis that high task demand drives auditory estimation down to the primary sensory area [21]. The right STS was only observed in imagined singing but not in listening conditions, which is consistent with previous findings [25], suggesting a possible specific functional role of STS in auditory imagery. The additional frontal (premotor) and parietal (TPJ, IPS) activations in imagined singing suggested that auditory representations, similar to the representations established in perception, can be constructed via the motor-to-sensory transformation pathway [3, 22, 25].

The observations of neural structures that operate at separate frequency bands are consistent with our hypothesis regarding the algorithms and neural implementations in the motor-to-sensory transformation. The phase coherence and its distribution during imagery are distinct from previous observations in speech perception [e.g. 29, 40, 41, 61, 62–64], suggesting unique motor-to-sensory transformation during mental operations. Specifically, first—using songs with a rhythm in the delta range—a complete frontal-parietal-temporal network was revealed in the corresponding low-frequency band during the imagined singing tasks (Fig 2D). These results provide functional evidence suggesting the extent of the motor-to-sensory transformation pathway, consistent with findings in neuroimaging studies [65–67]. More importantly, the neural phase tracking in the delta band suggested that the fluctuation of excitability in the network temporally aligned with the unfolding of mental operations.

Second, phase coherence at the theta band was localized at temporal auditory regions (Fig 3A). These results are consistent with the view that theta oscillations actively mediate the auditory and speech processes [40, 41]. In the context of motor-to-sensory transformation, the precise localization of the theta band suggested that theta oscillations could mediate specific computations for constructing speech-like representations.

Third, the mu and beta bands have indicated neural rhythmic signatures for both motor control and imagery [e.g., 42, 43]. In this study, we observed a broad sensorimotor network operating in the mu and beta rhythms during the imagined singing tasks (Fig 3B and 3D). These results further suggested that computations could be executed at these rhythms to internally simulate action during imagery [3, 12, 20, 22, 25]. Such motor simulations presumably generate an efference copy that is used to emulate the somatosensory consequences of actions for estimating the status of motor effectors [3, 22, 25].

The hemispheric lateralization suggests the possible commonalities and distinctions between inner speech and covert singing. The bilateral observations in the delta band supported neural entrainment to the common rhythmicity in both types of mental operations.

The lateralized theta band in the left auditory cortices was consistent with speech-weighted processing [68], whereas the right-hemispheric dominance in the mu and beta bands over the sensorimotor cortices suggested the potential construction of melodic and tonal information in the singing context [e.g., 44, 45, 46]. The switching from right to left hemisphere is consistent with the observations that the contents of auditory information determine the hemispheric lateralization—the right-lateralized processing of the humming tones is switched to left lateralization when the tones are superimposed on syllabic contents [69].

Fourth, the gamma band was observed over the entire motor and sensory systems. This result indicated the implementation of local computations in modality-specific regions [47, 48]. Together, the motor-to-sensory transformation network possesses distinct dynamics for mediating mental operations. The frontal-parietal-temporal network entrains the rhythm of metal operations. This low-frequency entrainment is suggested to facilitate communication and coordination between cortical areas for specific tasks via traveling waves [70–72]. Moreover, via cross-frequency coupling [47], specific frequencies (theta for auditory, mu and beta for sensorimotor) may modulate the gamma band to achieve the computations of motor simulation as well as somatosensory and auditory estimation in the motor-to-sensory transformation in the imagery tasks [48, 68]. Future studies are necessary to investigate these hypotheses regarding the connectivity across frequencies and cortical regions so that the functional contributions of the proposed model can be thoroughly examined.

This study also provides hints about the possible functions of neural oscillations on perception. Many studies have demonstrated that neural oscillations can entrain speech signals [28, 29]. However, it is still in debate whether the entrainment is driven by the stimuli features or is modulated by top-down factors on intrinsic oscillations [35, 73]. Previous neural entrainment experiments have used external stimuli and investigated how neural oscillations track physical features. The perceptual constructs are thus derived from the external stimuli features. It is hard, if not impossible, to separate perception from stimuli, and therefore the debate cannot be resolved. Our results of imagined singing without external stimulation suggest that the phase of neural oscillations can align with internal mental operations. That is, internally constructed representations can modulate the phase of neural oscillations during rhythmic mental imagery. These results support the view that top-down factors modulate intrinsic neural oscillations.

Our results may impact practical and clinical domains. The motor-to-sensory transformation network for imagined speech may implicate novel strategies for building a brain-computer interface (BCI). Previously, direct BCIs mostly focused on the motor system [74]. Our findings of synchronized neural activity across motor and sensory domains during mental imagery suggest possible updates for decoding algorithms from a system-level and multimodal perspective, which is hinted at in a recent advance [75]. Moreover, our results offer insights into the functional and anatomical foundations of auditory hallucinations. We have previously hypothesized that, from a cognitive perspective, auditory hallucinations may be caused by incorrect source monitoring of internally self-induced auditory representations [3, 12]. These results of synchronized neural activity in the frontal-parietal-temporal network suggest the possible neural pathways for the internal generation of auditory representation. These results are consistent with the neural modulation treatment for auditory hallucinations, which targets the motor-to-sensory transformation network with electric stimulation [76].

Using a rhythmic entrainment imagined singing paradigm, we observed that the neural phase was modulated at the rate of inner speech and covert singing. The synchronized activity spanned across dedicated frontal-parietal-temporal regions at multiple frequency bands, which is evidence of the motor-to-sensory transformation. The coherent activation in the

motor-to-sensory transformation network mediated the internal construction of perceptual representations and formed the neural computational foundation for mental operations.

## Materials and methods

### Ethics statement

The study was approved by the New York University Institutional Review Board (IRB# 10–7107) and conducted in conformity with the 45 Code of Federal Regulations (CFR) part 46 and the principles of the Belmont Report.

### Participants

Sixteen volunteers (7 males, mean age = 25 [19–32] years) participated in this experiment and received monetary compensation. All participants were right-handed, native English speakers, and without a history of neurological disorders. All participants provided informed written consent and received monetary compensation for their participation.

The number of participants was predetermined based on our previous studies that investigated similar speech imagery using MEG [21, 24]. Moreover, the number of 16 participants is in the upper range of the total participants used in previous MEG studies that investigated neural entrainment [e.g., 33, 34, 41, 77]. The effect size that is required to have 80% power at an alpha level of 0.05 for a sample size of 16 is Cohen's $d = 0.99$. As the standard deviation estimated in our study was 0.07, the absolute effect size in the measure (phase coherence) was 0.07. The post hoc effect size from our data in all sensors was $d = 1.50$ ($d_z = 2.32$). The absolute effect size is 0.09. In the clustering analysis using in this study, we used a pre-cluster threshold of 0.001 in the paired $t$ test. Considering the sample size of 16, the minimal effect size is $d_z = 1.02$. The post hoc effect size from the data in the HG that was independently defined by cortical parcellation (S1 Table and S1 Fig) was $d = 1.21$ ($d_z = 1.89$). The absolute effect size is 0.11.

### Materials

Female vocals were recorded singing the first sentence of 4 well-known songs (Alphabet Song: 6.24 s, 16 syllables, 2.56 syllables/s; Itsy Bitsy Spider: 7.38 s, 23 syllables, 3.12 syllables/s; Take Me out to the Ball Game: 5.42 s, 13 syllables, 2.40 syllables/s; and Twinkle Twinkle Little Star: 6.10 s, 14 syllables, 2.30 syllables/s). Arguably, the faster the songs were, the more difficult it would be to sing them at a similar speed across trials (e.g., more temporal variance). Therefore, we recorded the songs at a normal comfortable and intermediate speed (about 3 Hz of syllabic rate) to minimize the temporal variance and thus maximize the statistical power of neural entrainment in the imagery conditions. All songs were recorded with a sampling rate of 44.1 kHz. During the experiment, stimuli were normalized and delivered at about 70 dB SPL via plastic air tubes connected to foam earpieces (E-A-R Tone Gold 3A Insert earphones, Aearo Technologies Auditory Systems, Indianapolis, IN).

### Procedure

A fixation cross was presented in the center of the screen throughout the experiment. Color changes of the fixation were used as cues for the listening and imagery tasks to avoid contrast changes (onset and offset of the fixation) that could induce large visual responses. Participants were asked to listen to one of the 4 songs when the color of fixation was red (the listening condition). The fixation changed to yellow after the auditory stimulus offset. After 1.5 s, the fixation changed to purple, and participants were required to imagine singing the song that they just heard (the imagery condition). They were asked to covertly reproduce the song using the

same rhythm and speed as the preceding auditory stimuli. Participants pressed a button to indicate the completion of imagery. RT was recorded between the onset of the visual cue for imagery and when the button was pressed to indicate the completion of imagery. After the button press, the fixation turned yellow and stayed on screen for 1.5 s–2.5 s (with an increment of 0.333 s) until the next trial began. Participants were required to refrain from any overt movement and vocalization during imagined singing. A video camera and a microphone were used to monitor any overt movement and vocalization throughout the experiment.

Four blocks were included in this experiment, with 24 trials in each block (6 trials per song in each block, 24 trials per song in total). The presentation order was randomized. Participants were familiarized with the experimental procedure before the experiment.

## Behavioral analysis

The song Twinkle Twinkle Little Star was used for a research question independent from this study. Therefore, only 3 songs were used for further analysis. The RT was quantified as the duration between the onset of visual cue for imagery and participants' bottom press that indicated the end of imagery. The mean RT was obtained for the imagery of each song. The RT data were further transformed into z-scores. The distribution of z-scores was obtained and averaged across the 3 songs. The RT z-scores of 24 trials were ranked from shortest to longest for each song and averaged across the 3 songs. Two groups were formed based on RT ranking: the center group consisted of the 12 trials closest to the mean RT, whereas the dispersed group comprised the other 12 trials that were farther away from the mean RT. Because the separation of trials was defined by the differences between individual trial RT and the mean RT in the imagery condition, the RT differences among trials in the center group were much smaller than those among the trials in the dispersed group (Fig 1D). Similar durations indicated that trials in the center group were more likely imagined in a similar temporal manner. These temporal differences between groups of trials were used in the phase-coherence analysis of MEG to investigate our hypothesis of neural entrainment to imagery.

## MEG recording

Neuromagnetic signals were measured using a 157-channel whole-head axial gradiometer system (KIT, Kanazawa, Japan). Five electromagnetic coils were attached to each participant's head to monitor the head position during MEG recording. The locations of the coils were determined with respect to 3 anatomical landmarks (nasion, left and right preauricular points) on the scalp using 3D digitizer software (Source Signal Imaging, San Diego, CA) and digitizing hardware (Polhemus, Colchester, VT). The coils were localized to the MEG sensors at the beginning and the end of the experiment. The MEG data were acquired with a sampling frequency of 1,000 Hz, filtered online between 1 Hz and 200 Hz, with a notch at 60 Hz.

## MEG analysis

Raw data were noise reduced offline using the continuously adjusted least-squares method [78] in the MEG160 software (MEG Laboratory 2.001 M, Yokogawa Corporation, Eagle Technology Corporation, Kanazawa Institute of Technology). We used independent component analysis (ICA) to reject artifacts caused by eye movement and cardiac activity. Epochs were extracted for trials in the listening and imagery conditions, with each epoch of 6,000 ms in duration (including 2,000 ms pre-stimulus and 4,000 ms post-stimulus period). For the listening condition, 24 trials of each song were grouped and formed 3 within-groups. Furthermore, 8 trials were randomly sampled from 24 trials of each song and yielded a new group of 24 trials (between-group). This sampling procedure was conducted 3 times to form 3 between-groups.

The sampling was without replacement such that each trial was used only once in the 3 between-groups.

For imagery conditions, MEG trials were separated into center groups and dispersed groups for each song according to RT z-scores (see aforementioned behavioral analysis). The group separation in the imagery condition was different from the creation of groups in the listening condition. The comparison between the center group and the dispersed group overcame difficulties that were induced by the unique, active nature of imagery tasks. Unlike the fixed duration and dynamics in every listening trial, imagery performance varied from trial to trial. Based on established phase-coherence calculations using all trials, the imagery would reduce the statistical power that was already smaller than perception. The split-group analysis was an extension of the phase-coherence analysis by adapting to the unique requirement of imagery. Another option was to use an intrinsic baseline (rest), but this could introduce additional confounding factors, whereas trials in the dispersed group had identical procedures and tasks. The only difference between trials in the dispersed group and center group was the response timing that reflects the measure of interest (the dynamic processes during imagery). Therefore, the dispersed group was a better-controlled baseline. Using the temporal differences while maintaining everything else between imagery groups can precisely reveal the dynamics of interest in the mental imagery tasks using the analysis of phase coherence across trials.

A fast Fourier transform (FFT) was applied to each trial in a group with a 500-ms time window in steps of 200 ms, yielding 19 time points in each 4-s-long trial epoch. The phase values were extracted at each time point and frequency (1–48 Hz with a spectral resolution of 1 Hz). The ITC was calculated as Eq 1 [29]. In Eq 1, $\theta_j\,(t, f)$ represents the phase value at time point $t$ and frequency $f$ in the $j$th trial. $N$ represents the total number of trials in a group. The ITC values were obtained for each of the within-groups and between-groups in the listening condition, and each of the center groups and dispersed groups in the imagery condition.

$$ITC(t,f) = \left(\frac{\sum_{j=1}^{N} cos\ \theta_j(t,f)}{N}\right)^2 + \left(\frac{\sum_{j=1}^{N} sin\ \theta_j(t,f)}{N}\right)^2 \tag{1}$$

The ITC characterized the consistency of the temporal (phase) neural responses across trials. If the phase responses were identical across trials, the ITC value would be 1. To investigate the neural entrainment to the rhythms in the external stimuli and the mental operations, the ITC values were first averaged in the delta band (1–3 Hz). According to our hypothesis that neural responses track the rhythm in acoustic signals and imagery at the syllabic rate of 2–3 Hz, similar durations among trials in the center group should yield higher ITC values than those from trials in the dispersed group in the imagery condition, and trials in the within-group should yield higher ITC values than those from trials in the between-group in the listening condition.

Next, the ITC values were averaged over time (0–4 s) and across the 3 songs to yield a single value in every MEG channel for the within-group and between-group in the listening condition, and the center group and dispersed group in the imagery condition. For the between-group in the listening condition, the random grouping procedure was repeated 100 times and yielded 100 ITC values. The 50th percentile of ITC was chosen as a baseline to compare with the ITC of the within-group in the next cluster-based analysis.

Distributed source localization of ITC was obtained by using Brainstorm software [79]. The cortical surface was reconstructed from individual structural MRI using Freesurfer (Martinos Center for Biomedical Imaging, Massachusetts General Hospital, Boston, MA). Current sources were represented by 15,002 vertices. The overlapping spheres method was used to compute the individual forward model [79]. The inverse solution was calculated by

approximating the spatiotemporal activity distribution that best explained the ITC value. Dynamic statistical parametric mapping (dSPM) [80] was calculated using the noise covariance matrix estimated with the 1,000-ms pre-stimulus period. To compute and visualize the group results, each participant's cortical surface was inflated and flattened [81] and morphed to a grand average surface [82]. Source data were spatially smoothed using a Gaussian smoothing function "SurStatSmooth" in the SurStat toolbox [83] with 3 mm of Full Width at Half Maximum (FWHM).

The nonparametric cluster-based permutation test [84] was used to assess significant differences between groups in the source space [85]. For the listening condition, the ITC values of the within-group were compared with the baseline ITC values of the between-group. For the imagery condition, the ITC values of the center group were compared with the dispersed group. The empirical statistics were first obtained by a two-tailed paired $t$ test with 2 or more adjacent significant vertices with a pre-cluster threshold of alpha = 0.001. Next, a null distribution was formed by randomly shuffling the group labels 1,000 times. Cluster-level, FDR-corrected results were obtained by comparing the empirical statistics with the null distribution (cluster threshold of alpha = 0.05 for both listening and hearing conditions).

To test the possible differences between listening and imagery, we directly compared the activations in the listening and imagery conditions. The ITC values in the within-group of the listening condition were compared with the ITC values in the center group of the imagery condition using the same nonparametric cluster-based permutation test. Moreover, to test the possible common cortical regions that mediate both listening and imagery, we conducted a conjunction analysis. The cortical regions that had significant main effects in both listening and imagery conditions were identified and depicted on the cortical surface.

To further explore the functional specificity of dynamic processing in the motor-to-sensory pathway, we tested the phase coherence across trials in 5 more frequency bands—the theta (4–8 Hz), alpha (9–12 Hz), low-beta (13–16 Hz), mid-beta (17–20 Hz), high-beta (21–28 Hz), and low-gramma (30–48 Hz) bands. Similar procedures were implemented as those used in the investigation in the delta band, except the phase values were extracted at corresponding frequencies. The phase coherence in each frequency band was obtained using Eq 1. The same distributed source localization and nonparametric cluster-based permutation tests were applied to each frequency band.

To further test the reliability of the results, we used a long epoch in the phase-coherence analysis. The duration of imagery across trials and among participants could vary over a wide range. The imagery durations in some trials, especially imagery of the shortest song, could be shorter than the length of epochs if long epochs were extracted. To be conservative, we first chose a stringent threshold of 4 s. It is possible that more data points may balance the reduction of statistical power that is due to the variance of imagery durations across trials. Therefore, we applied FFT on 10-s-long epochs (2 s of pre-stimulus and 8 s of post-stimulus) and applied the same phase-coherence analysis to the 6-s post-stimulus data.

To probe the evolution of phase coherence across time, we performed an analysis on shorter, consecutive time bins. The 6-s post-stimulus data were binned into four 3-s-long time bins, with 2-s overlaps between 2 consecutive time bins (0–3 s, 1–4 s, 2–5 s, and 3–6 s). The data in each time bin were subject to the same phase-coherence analysis, followed by the same distributed source localization and nonparametric cluster-based permutation tests.

## Supporting information

**S1 Fig. Cortical parcellation of a grand average surface across 16 participants.** Parcellation superimposed on the inflated average cortical surface. Cortical areas related to this study were

labeled with numbers on the left hemisphere. Refer to the S1 Table for the anatomical names for labels. The numeric labels were consistent with the ones used by Destrieux and colleagues [86].
(TIF)

**S2 Fig. Phase-coherence results using 6-s-long epochs.** The results for (A) the listening condition and (B) the imagery condition were consistent with the results using 4-s-long epochs in Fig 2C and 2D. The underlying data for this figure can be found at https://osf.io/mc8wd/.
(TIF)

**S3 Fig. Evolution of phase coherence in four 3-s time bins.** The results were more reliably toward the end of trials and were consistent with the results in Fig 2D. The underlying data for this figure can be found at https://osf.io/mc8wd/.
(TIF)

**S1 Table. Cortical parcellation and labels, used in S1 Fig.**
(DOCX)

## Acknowledgments

We thank Jeff Walker for his technical support in MEG recording, Ling Liu, Gregory Cogan, Gaoxing Zheng, and Xiaoxuan Wang for their help on data analysis and source localization, and Yi Jie (E'jane) Li and Anna Zhen for commenting on an earlier draft.

## Author Contributions

**Conceptualization:** Huan Luo, Xing Tian.

**Data curation:** Xing Tian.

**Formal analysis:** Yanzhu Li.

**Supervision:** Xing Tian.

**Writing – original draft:** Yanzhu Li, Xing Tian.

**Writing – review & editing:** Yanzhu Li, Huan Luo, Xing Tian.

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
