## [Editor Report · Decision Letter 0]

10 Sep 2019

Dear Dr Tian, 

Thank you for submitting your manuscript entitled "Verbal thinking in rhythm: motor-to-sensory transformation network mediates imagined singing" for consideration as a Short Report by PLOS Biology. Please accept my apologies for the delay in sending this initial decision to you. We were interested in your study, and thus, sought advice from an Academic Editor with relevant expertise. With that advice now in hand, I’m pleased to let you know that we would like to send your submission out for external peer review.

Please re-submit your manuscript within two working days, i.e. by Sep 12 2019 11:59PM.

Kind regards,

Gabriel Gasque, Ph.D.,

Senior Editor

PLOS Biology

---

## [Decision Letter · Decision Letter 1]

13 Nov 2019

Dear Dr Tian,

Thank you very much for submitting your manuscript "Verbal thinking in rhythm: motor-to-sensory transformation network mediates imagined singing" for consideration as a Short Reports at PLOS Biology. Your manuscript has been evaluated by the PLOS Biology editors, by an Academic Editor with relevant expertise, and by three independent reviewers. Please accept my sincere apologies for the long delay in sending the decision below to you.

As you will see, while the reviewers agree the study is mostly technically solid, there is disagreement about the novelty of the findings. More importantly, however, is the fact that reviewer 3 has identified the low number of participants as a major problem; something that shouldn’t “be advocated, particularly in a wide-reaching journal like Plos Biology”. Having discussed this specific concern with the Academic Editor, we have decided to decline further consideration.

However, we would be willing to consider a heavily revised manuscript that addresses all the reviewers’ comments, particularly by increasing the number of volunteers. Please note that by leaving the door open for re-submission, we have decided that your findings, if confirmed by a larger sample size, offer the novelty that we aim to publish in our journal.

In addition, our Academic Editor agrees with reviewer 1, especially on the analysis of other frequency bands. You should reframe the “inner thinking” concept with imaging the rhythmicity and tonality of a song, as suggested by reviewer 3, and the direct statistical comparison between the Listening and Imagery groups is a must.

We appreciate that these requests represent a great deal of extra work, and we are willing to relax our standard revision time to allow you six months to revise your manuscript. Please email us (plosbiology@plos.org) to discuss this if you have any questions or concerns, or think that you would need longer than this. At this stage, your manuscript remains formally under active consideration at our journal; please notify us by email if you do not wish to submit a revision and instead wish to pursue publication elsewhere, so that we may end consideration of the manuscript at PLOS Biology.

Your revisions should address the specific points made by each reviewer. Please submit a file detailing your responses to the editorial requests and a point-by-point response to all of the reviewers' comments that indicates the changes you have made to the manuscript. In addition to a clean copy of the manuscript, please upload a 'track-changes' version of your manuscript that specifies the edits made. This should be uploaded as a "Related" file type. You should also cite any additional relevant literature that has been published since the original submission and mention any additional citations in your response. 

Before you revise your manuscript, please review the following PLOS policy and formatting requirements checklist PDF: http://journals.plos.org/plosbiology/s/file?id=9411/plos-biology-formatting-checklist.pdf. It is helpful if you format your revision according to our requirements - should your paper subsequently be accepted, this will save time at the acceptance stage.

Please note that as a condition of publication PLOS' data policy (http://journals.plos.org/plosbiology/s/data-availability) requires that you make available all data used to draw the conclusions arrived at in your manuscript. If you have not already done so, you must include any data used in your manuscript either in appropriate repositories, within the body of the manuscript, or as supporting information (N.B. this includes any numerical values that were used to generate graphs, histograms etc.). For an example see here: http://www.plosbiology.org/article/info%3Adoi%2F10.1371%2Fjournal.pbio.1001908#s5.

For manuscripts submitted on or after 1st July 2019, we require the original, uncropped and minimally adjusted images supporting all blot and gel results reported in an article's figures or Supporting Information files. We will require these files before a manuscript can be accepted so please prepare them now, if you have not already uploaded them. Please carefully read our guidelines for how to prepare and upload this data: https://journals.plos.org/plosbiology/s/figures#loc-blot-and-gel-reporting-requirements.

Upon resubmission, the editors will assess your revision and if the editors and Academic Editor feel that the revised manuscript remains appropriate for the journal, we will send the manuscript for re-review. We aim to consult the same Academic Editor and reviewers for revised manuscripts but may consult others if needed.

If you still intend to submit a revised version of your manuscript, please go to https://www.editorialmanager.com/pbiology/ and log in as an Author. Click the link labelled 'Submissions Needing Revision' where you will find your submission record. 

Sincerely,

Gabriel Gasque, Ph.D., 

Senior Editor

PLOS Biology

Reviewer remarks:

Reviewer's Responses to Questions

Reviewer #1: The study by Li and colleagues investigated whether imagined singing was associated with activation of the motor-to-sensory-transformation network, as quantified by the M100 response and changes in phase coherence in the delta frequency band (which corresponded to the approximate rhythm of the song frequency). They reported that imagined singing produced changes in measures that were comparable (in pattern and magnitude) to those elicited by passive listening to the songs. This study follows on from the team’s influential work on the role of the motor-to-sensory transformation network in inner speech. The design was creative and innovative; I particularly liked the idea of frequency tagging the rhythm of inner speech (discussed further below). 

I have some questions and comments regarding the methodology and results that the authors could consider: 

- If I could just clarify my understanding of the experimental design: participants were first played the full song (~ 5 – 8 seconds – these constituted the ‘listening trials’). There was then a gap of 1.5 seconds. Then the fixation cross changed color and participants were required to imagine themselves singing the song (these constituted the ‘imagery trials’). Is this correct? 

- The similarity in the form and magnitude of the M100 elicited by the song onset in the listening condition and the imagery onset in the imagery condition is impressive, and the authors’ claim that the auditory cortices were activated in the imagery conditions seems plausible (to me). However, is it possible that this component was (partly) elicited by the change in color of the fixation cross? It would presumably be easy to discount this possibility by measuring the MEG response to a colour change presented alone (i.e., in the absence of auditory stimulation or song imagery). 

- Regarding the temporal domain (waveform) analysis: I don’t understand why only the first sound (or sound image) of the song elicited an M100 – can the authors clarify? If this signal truly reflects the activation of the auditory cortex by the imagined song, wouldn’t each individual letter be expected to elicit an M100? If might be useful to illustrate the onset of each individual letter in the listening condition (e.g., by means of a dotted line), to see if each letter is followed by a M100 (albeit smaller than the M100 elicited by the first letter). Relatedly, in the Results the authors state: “Moreover, no repetitive patterns were observed in the time course of listening or imagery (Fig. 1g&h), suggesting that the tracking of the acoustic stream or the rate of imagery was not by the response magnitude” – as above, I don’t understand why the authors were not expecting a repetitive pattern if each specific ‘sound image’ was associated with an M100, and particularly given that they claim to have identified delta oscillations to the inner song. Can the authors clarify why they would expect the first sound image in the sound to elicit an M100, but not subsequent sound images? An alternative explanation is that that the signal actually reflects an ‘omission M100’ response (akin to the oN1 response in the ERP literature) caused by a violation in the expectation of hearing a sound – this could perhaps explain why the signal is only present to the first sound in the song. I would be interested to hear the authors’ thoughts on this. 

- Regarding the time-frequency (spectral) analysis: I think it would be helpful if the authors could add plots showing the change in phase coherence across time, and across a range of frequencies (i.e., demonstrate that the inner singing elicited oscillations). Without this information, how can the authors be sure that the ‘entrainment’ they report is specific to the delta band (i.e., the approximate frequency of the rhythm)? While their hypothesis is plausible, it seems to me first necessary to show that the entrainment is specific to delta, and does not also occur in other frequency bands (theta, alpha, beta, gamma, etc.). 

- I thought the ‘frequency-tagging’ aspect of the design was interesting and creative. However, given that the rhythm was approximately equivalent for all of the songs (and the data were averaged across songs in any case) I feel like this aspect of the design could be developed further. For example, it would be interesting to compare the oscillations elicited by ‘delta rhythm’ songs vs. ‘theta rhythm’ songs, to see if the oscillations they elicit are specific to their rhythm: a sort of ‘auditory steady-state response’ for imagined sounds. 

- I’m not sure I fully understand the rationale for comparing the ‘center-group’ (which would more accurately be labelled as ‘center-trials’) and the ‘dispersed-group’ in the imagined song analysis – is the idea that the motor-based predictions are better in the ‘center-trials’, as the RTs better matched the duration of the actual song? If so, I’m not sure I agree – given that participants overestimated the duration of the song in all imagery conditions, doesn’t this imply that the trials with the lowest RTs actually reflect the most accurate predictions? But in any case, why would the ‘motor-based-predictions’ be expected to be stronger in the center trials? What is the benefit of this approach against, say, simply comparing the ITC when (a) participants listened to songs, (b) participants produced inner songs, and (c) participants sat passively. Presumably the (b) vs. (c) comparison would be expected to be even stronger in such an analysis? 

- With regards to the source-localization: why did the critical p-values differ between the listening and imagery conditions – did the number of statistical comparisons differ between these conditions? As an aside, while the required number of trials per condition is a matter of debate, 12 trials per condition seems pretty minimal for a paradigm with no external stimulation where a large effect size would presumably not be expected – can the authors comment?

Reviewer #2: In « Verbal thinking in rhythm: motor-to-sensory transformation network mediates imagined singing », Li and colleagues aimed to define the brain networks associated to imagined singing with an elegant design and methodological approach (frequency tagging on MEG data). 

The study is interesting and rigorously performed using state-of-the art analyses tools. However I am wondering about the novelty and significance of the findings, as the implication of fronto-temporo-parietal networks in speech and music mental imagery is already very well established (as pointed out by the authors in the introduction and discussion sections). Moreover the potential novelty of this study, as stated by the authors (naturalistic sounds, frequency tagging approach, and MEG source localization), is mainly methodological and is, in my opinion, not so novel because these approaches have been used by plenty of research groups in the domain. 

In its current form, I thus think if this report might be more suitable for a more specialized journal.

I only have minors comments regarding the methodology and the main text that I list below :

- The term Reaction Time (RT) is confusing, the authors could consider using a more explicit term that state that this metric meaure the duration of imagination period

- In a related vein, the sentence « We found that when participants

imagined with less temporal variation … » can be reformulated in order to clarify that it concerns the variation of the duration of the imagination period accross trials 

- Could the authors justify why they used ICA instead of Signal Space Projection (SSP) for correction of eye movement and cardiac artifacts? 

- « Furthermore, eight trails were randomly sampled from 24 trials of each song and yielded a new group of 24 trials (between-group) » : change trails by trials

- “Comparing the phase coherence results in imagined singing with those in listening conditions, the observations were overlapped in the temporal regions of primary and secondary auditory cortices. » The authors did not compare the listening and imagination conditions and did not estimate the conjunction (overlap) between these conditions. These analyses should be added if the authors want to make this statement.

Reviewer #3: Li and colleagues use a clever design to study where in the brain we can find phase-synchronised activity while participants imagine to sing a few lines of children’s songs. Significant phase coherence in the delta band can be found in a large network comprising bilateral inferior frontal, motor, temporal, and parietal areas. The analysis is new in the context of imagined singing (though standard for speech studies). The study has the potential to inform about the neural representation of imagined singing, but I have a few major (essential) and minor concerns and questions.

Major

1. The study is framed in a “verbal thinking” or “inner speech” context (as indicated by title, abstract, introduction and discussion), but I don’t think it is appropriate to equate thinking with singing. Singing (production and perception) involves a wider/different network than thinking. The right hemisphere for example, is much more involved in singing than verbal thinking. In my opinion, the manuscript needs to reflect these basic differences and the topic needs to be reframed. 

2. Related to 1: I imagine that the same pattern of results would arise if participants were to just imagine humming the song, keeping the rhythmicity. I don’t think the results necessarily show processes related to “inner thinking” or speech, but they show consistent neural processes related to the rhythmicity and tonality of imagining a song. This should at least be discussed.

3. The phase synchronisation results during Listening and Imagery are compared in the discussion, but a statistical comparison in the results is missing. This is the most important contrast in this study (to find out what the “inner” aspect of singing actually adds/changes when compared to simply listening to songs). This could for example be done on a trial-by-trial basis.

4. Related to the previous point – there were two different alpha levels used for analysis in the Listening and Imagery conditions, so their results can’t be compared. I.e. just before the result section: “(alpha=0.05 for the listening conditions, and alpha=0.001 for the imagery conditions).” In the result section itself, there’s even another alpha level mentioned: “For the imagery condition, [..] (pcorr(FDR)<0.01)”. The same thresholds should be used for all conditions, unless there is a clear justification for this.

5. The study had only 16 participants (and no a priori estimated sample size), which is becoming less and less acceptable in psychological and neuroimaging studies (e.g. Simmons JP, Nelson LD, Simonsohn U (2011) False-positive psychology: Undisclosed flexibility in data collection and analysis allows presenting anything as significant. Psychological science 22: 1359-1366. https://doi.org/10.1177/0956797611417632). While it would be unfair to request additional data, a sample size of 16 is not something that should be advocated, particularly in a wide-reaching journal like Plos Biology. Please add an explicit justification for this low number of participants to demonstrate awareness that this is not good scientific practice.

6. The authors propose a model (Figure 1f) that describes Articulatory Simulation, Articulatory Estimation, and Auditory Estimation, including three different areas involved in these processes. While the results support that these areas (and others) show increased delta phase coherence, none of the three proposed processes can be disentangled, using the present analysis. It might be possible to gain support for some aspects of the model using additional analyses. For example, using directed connectivity, the proposed direction of information flow could be supported. But the present paradigm cannot provide support for the different processes of the model. Please make it clear in the manuscript that the used approach is not suitable to study the different functional contributions of the proposed model (only the involvement of the areas in delta phase alignment).

Minor

1. The language in the manuscript is problematic. I understand that it can be extremely difficult as a non-native speaker to write in English and reviewers should as much as possible try to ignore this. But in the current manuscript it is often unclear whether a statement refers to previous research, the current study, or something that is proposed here (due to the unclear use of tenses). Please ask a native speaker to proofread the ms before re-submission.

2. Please add the fourth song to the analysis, even if is also used for a different research question. With only 16 participants, this study needs as much data as possible.

3. In the introduction, particularly on page 3, a lot of information is missing at this point to understand the statements here. It is unclear what is meant with “reaction time” or what the conditions are. The descriptions here don’t make sense without knowing the paradigm. Please explain briefly what the participants had to do (maybe at the beginning of the last paragraph of the intro).

4. Please add page and line numbers to the manuscript.

5. The approach used here is described as “frequency tagging”. Frequency tagging is well defined in the field and I think neither the listening nor (even less) the imagery condition can be classed as frequency tagging.

6. In the figure, please indicate the areas in the result plots (1i and 1j) that refer to proposed areas in 1f, to make a comparison of hypotheses and results easier.

7. Please use a more informative plot for 1b. Bar plots are becoming unacceptable in scientific research.

8. Source data were spatially smoothed – using which parameters?

9. Please define the terms of the equation (i.e. t,f,N,θ). Should the θ be δ?

10. Why were only 4 s post-stimulus analysed? The songs were a minimum of 5.42 s long, and participants needed consistently longer to imagine the songs. I would think that 6 s post-stimulus would maximise available data for each epoch, and I would also add an additional 2 s (consistent with the pre-stimulus time) to optimise the FFT results (leading to 10 s [-2s to 8 s] epochs).

---

## [Decision Letter · Decision Letter 2]

13 Aug 2020

Dear Dr Tian,

Thank you for submitting your revised Short Report entitled "Mental operations in rhythm: motor-to-sensory transformation mediates imagined singing" for publication in PLOS Biology. I have now obtained advice from the original reviewers and have discussed their comments with the Academic Editor. You will note that reviewer 1, Thomas Whitford, has identified himself. 

Based on the reviews, we will probably accept this manuscript for publication, assuming that you will modify the manuscript to address the remaining points raised by the reviewers. Please also make sure to address the data and other policy-related requests noted at the end of this email.

We expect to receive your revised manuscript within two weeks. 

**IMPORTANT:

Your revisions should address the specific points made by each reviewer. However, we will not press for the inclusion of additional analyses.

In addition, please remove the following paragraph from your manuscript, because our expert on statistical analyses thinks it is a circular argument:

"The post-hoc effect sizes obtained in multiple analyses are greater than the required effect size. Therefore, using the predetermined sample size of 16 that is consistent with previous studies should provide enough statistical power to reliably measure the well-documented neural entrainment effects using MEG."

Please submit the following files along with your revised manuscript:

In addition to the remaining revisions and before we will be able to formally accept your manuscript and consider it "in press", we also need to ensure that your article conforms to our guidelines. A member of our team will be in touch shortly with a set of requests. As we can't proceed until these requirements are met, your swift response will help prevent delays to publication.

*Copyediting*

*Published Peer Review History*

*Early Version*

*Submitting Your Revision*

Sincerely,

Gabriel Gasque, Ph.D.,

Senior Editor,

ggasque@plos.org,

PLOS Biology

ETHICS STATEMENT:

-- Please indicate within your manuscript whether your protocols approved by the Institutional Review Board (IRB) at New York University adhered to the Declaration of Helsinki or any other national or international ethical guidelines.

-- Please include the ID number of the protocol approved by the Institutional Review Board (IRB) at New York University

-- Please indicate if participants gave informed written consent. If consent was oral, please explain why.

DATA POLICY:

-- Please include in our deposition in OSF a README file that would allow the reader to link your data files to each of the figures displaying quantitative data, by explaining how the data was analyzed to generate the final plots and graphs.

-- In addition to you raw MEG data, please upload or provide as a supporting file a spreadsheet containing the individual numerical values that were used to generate the summary statistics show in figures 1AB. For an example see here: http://www.plosbiology.org/article/info%3Adoi%2F10.1371%2Fjournal.pbio.1001908#s5

Reviewer remarks:

Reviewer #1, Thomas Whitford: The authors have considered my comments carefully and made detailed responses. I have no further comments.

Reviewer #2: I thank the authors for this revised manuscript and for taking my comments into account. The article has been greatly improved.

 I have one final comment concerning the last analysis: The ITC for the theta, mu, beta and gamma bands has been computed only for the imaginary condition: in order to be able to conclude that these different oscillatory dynamics (and networks) are specific to mental imagery a direct contrast between the imaginary and listening condition (for each frequency band) is needed.

Reviewer #3: I would like to thank the authors for their thorough reply and the additional analyses, in particular the statistical contrast between conditions and the conjunction plot. I also like the inclusion of the other frequency bands. Overall, my comments have been adequately addressed. 

The only thing I would like to add after reading the revision is that I would be very careful with the idea that the current study can highlight "commonalities and distinctions between inner speech and covert singing" (highlighted manuscript: page 25/line 3, clean manuscript: page 24/line 22). It is true that nursery rhymes are somewhere in between these two stimulus types, but I don't think that it can highlight differences and commonalities between inner speech and covert singing. To do this, it would be necessary to use these two types of stimuli/tasks separately. Any conclusions regarding the differences/similarities based on this used stimulus type must remain speculative (which is okay in a discussion). Also, these conclusions use "reverse-inferring" based on previously found brain regions and this is usually selective and not necessarily informative or applicable. Could the authors express their thoughts on page 25 more carefully and explicitly highlight that this is speculative?

---

## [Editor Report · Decision Letter 3]

1 Sep 2020

Dear Dr Tian,

On behalf of my colleagues and the Academic Editor, Hugo Merchant, I am pleased to inform you that we will be delighted to publish your Short Reports in PLOS Biology. 

Early Version

PRESS 

Kind regards,

Alice Musson

Publishing Editor, 

PLOS Biology

on behalf of

Gabriel Gasque,

Senior Editor

PLOS Biology